# The Role of Knowledge Creation and Transfer in Family Firm Succession

Kalin Pipatanantakurn and Vichita Vathanophas Ractham *

College of Management, Mahidol University, Bangkok 10400, Thailand; karin.pipat@gmail.com
* Correspondence: vichita.rac@mahidol.ac.th

**Abstract:** The purpose of the study is to investigate the role that knowledge creation and knowledge transfer processes play in family firm intergenerational succession in Thailand. An exploratory qualitative case study approach is used. Interviews were conducted with successors and predecessors of small, medium and large Thai firms that have undergone leadership succession within the past five years (30 firms, for $n = 60$ interviews). Data were analyzed using a qualitative content analysis approach. There were 16 different knowledge approaches identified that are undertaken by the successor. These processes are commonplace to firms, including formal and informal, internal and external processes of knowledge creation and transfer. Most of these occur at different stages of preparation for succession (pre-succession, transition and succession stages). While some knowledge approaches are used across firms, others are specific to small or large firms. These knowledge approaches and stages were used to develop a knowledge process model for family firm succession. The research develops an original model of the knowledge processes associated with family firm succession. This model, which incorporates a staged succession model with the knowledge processes identified, explains how and why knowledge creation and transfer occur during the succession process.

**Keywords:** succession planning; family firms; intergenerational knowledge transfer; succession knowledge transfer process model





## 1. Introduction

On a global scale, family firms are a dominant sector of the economy. One recent estimate shows that family firms account for over half of global GDP and around 67% of global employment [1]. During the COVID-19 pandemic, family firms have been highly resilient, with 64% expecting revenues to increase in 2021 and 79% not needing additional capital in 2020 [1]. Family firms are equally important in Thailand, where they comprise at least 80% of Thai firms, including around 75% of the Stock Exchange of Thailand (SET)-listed domestic firms, accounting for 64.9% of the total market valuation [2]. However, relatively little is known about some aspects of family firm management, including the transfer of knowledge during the leadership succession process. Family firms are characterized by the transfer of both knowledge and leadership between generations [3], but firms still suffer from change resistance and poor preparedness [1]. Some family firms may not even engage in active succession planning, or leave it too late to prepare effectively for the sudden death or incapacitation of the current leader [4]. Only around 30% of family firms successfully transfer between the first and second generations; later generational successions occur even less frequently [5]. Thus, the success of the succession process poses a major challenge for long-lived family firms.

One of the ways that the succession process can succeed is ensuring that the designated successor has the requisite knowledge to effectively lead the firm [4]. However, problems such as a reliance on tacit knowledge [6] and the multiple roles of leaders [7] make knowledge transfer particularly challenging. To date, there has been no formal investigation of knowledge processes and how they influence the knowledge creation and

transfer process throughout the succession cycle. This research gap limits both the practical understanding of succession preparation and planning within the family firm and the theoretical understanding of the role of knowledge transfer in the succession process.

The objective of this research is to investigate the role that knowledge creation and knowledge transfer processes play in family firm intergenerational succession in Thailand. The research questions include: (1) What processes are used for knowledge transfer in the succession planning of family firms? (2) When are these processes used during the succession process? (3) What differences can be observed between firms of different sizes? In answering these research questions, the researcher will be able to develop a theoretical model of knowledge transfer along the succession process lifecycle, beginning with the earliest stages of preparation (typically before the successor enters the workplace formally) and ending at the point of succession. This will be both a novel theoretical model and one that offers practical value for the succession planning process.

The paper begins with a review of the literature, including key concepts of the knowledge management process, knowledge management in family firms, succession in family firms, and the role of knowledge management in the succession process. This culminates with a presentation of the theoretical framework. The research method is then described. The findings resulting from the study are presented and then discussed. The final section of the paper is the conclusion, which summarizes the academic and managerial implications, limitations, and further research opportunities.

## 2. Literature Review

### 2.1. The Knowledge Management Process

The knowledge management process model that is most commonly used is the SECI model of knowledge transfer [6,8–10]. This model, which was developed by researchers Nonaka and Takeuchi [10] through a study of Japanese manufacturing firms, proposes a spiral model through which knowledge is transformed between tacit and explicit knowledge. Tacit knowledge is that which is individual, contextual and difficult to formalize, instead passing between individuals or being learned through physical experience [11]. Explicit knowledge, on the other hand, is shared and formalized, typically written down for future use and transfer between individuals [11]. The SECI model proposes four processes of knowledge transformation, which are: socialization (transfer of tacit knowledge between individuals); externalization (transformation of tacit knowledge to explicit knowledge); combination (creation of new knowledge through integration of two streams of explicit knowledge); and internalization (the integration of explicit knowledge into the individual's tacit knowledge) [6,8–10].

A key aspect of the knowledge management process is that knowledge is constructed through social interaction processes [12]. This research focuses on two specific knowledge construction processes. The first is knowledge creation, or social interaction intended to produce new knowledge (consistent with the Combination process of the SECI model) [13]. As Gourlay notes, the SECI definition of this process is problematic because it is limited to managerial interaction. Regardless, knowledge creation is central to organizational activities such as innovation [14]. Therefore, it is important for the organization's strategy. The second process of knowledge construction considered here is knowledge transfer, which is the direct or indirect communication of knowledge from one individual to another [15]. Unlike knowledge creation, knowledge transfer is not inherently a competitive advantage, but it does create the possibility that knowledge may be applied more widely; thus, it can, if used effectively, also constitute a competitive advantage.

### 2.2. Knowledge Management in Family Firms

There are several conditions of family firms that influence the knowledge management process. One of these characteristics is that family firms, regardless of size, have some degree of nepotism (or favorable treatment of family members) [16]. The effects of nepotism are complicated. In cases of entitlement nepotism (hiring and preferential treatment regard-

less of capability), nepotism can threaten the performance and sustainability of the firm. However, reciprocal nepotism (creation of reciprocal networks of support and obligation) can actually benefit the firm and enable the transfer of tacit knowledge [16]. Another characteristic of knowledge management in family firms is that they are often bound to tradition to a greater extent than non-family firms [17]. This does offer the family firm the opportunity to create and transfer knowledge drawn from this tradition (which Magistretti et al. term 'innovation through tradition'). However, it could also stymy knowledge creation and transfer due to change resistance, which can accumulate in tradition-bound firms [18]. Knowledge management in family firms is also strongly influenced by affective relationships and trust between the family members [19]. These affective bonds and trust, along with the family's history, influence the extent to which knowledge transfer will be successful, because they have an effect on communication patterns. The internal relationships of the family also have an effect on family firm culture [20]. The culture of family firms is established internally through family leadership, but it is also influenced through the relationships and interactions of the family members themselves. Thus, knowledge sharing and knowledge management in general are influenced by family relationships, not only the intergenerational knowledge transfer between family members [20]. Thus, knowledge management may be different in family firms than in non-family firms, due to the entanglement and 'familyness' of the relationship of members of the organization.

Additionally, family firms of different sizes may have different knowledge management processes, due to differences in resources, capabilities and absorptive capacity. Small and medium-sized enterprises (SMEs) have resource constraints, including financial and knowledge capital constraints, that limit the extent to which they can use formal knowledge management processes [21]. This means that SMEs may rely on socialization processes (for example on-the-job training), which transfer tacit knowledge directly between individuals, rather than a more formalized process of knowledge externalization or combination. They may also rely heavily on external sources of knowledge, including informal and formal sources; for example, they may draw on community connections, make use of external training and experts, or exploit available knowledge from competitors, government agencies and other stakeholders. This contrasts to larger firms, where internal knowledge and innovation is viewed as a competitive advantage [21]. Organizational resources also have a direct effect on knowledge management process capability development, meaning that resource-constrained firms may struggle to make use of the knowledge they do have [22]. Small firms may have not just resource constraints, but constraints on absorptive capacity [23]. These absorptive capacity constraints are important because, as the research of Chaudhary and Batra shows, absorptive capacity is directly related to the firm's entrepreneurial, market and technology orientations and, in turn, the firm's financial performance. Thus, small firms, including family firms, have resource, capability and absorptive capacity constraints that mean that knowledge management and its effects are not the same for these firms as for larger firms. In response to this, the present study anticipates that knowledge management processes will be different in small and medium firms compared to large firms.

### 2.3. Succession in Family Firms

In family firms, succession can be the period in the firm's lifecycle where managerial control and economic ownership of the firm are transferred to the younger generation, typically to a designated successor or successors who have been prepared for their new roles over a period of time [4]. The succession stage of the family firm's lifecycle is what makes the firm a sustained family firm [3]. This period can be a time of considerable strain in the family firm, both because of the need to prepare the successor and because of the organizational and familial tensions that arise [24]. These tensions can cause resistance to transfer of power, as well as social and organizational issues [25]. If not effectively managed through the succession process, this can cause the firm to fail, either immediately or over

time [26]. Thus, the knowledge creation and transfer processes used in the succession process are potentially critical for the effective firm succession.

This research uses Handler's succession process model, which is a role-based model [24,27]. In this four-phase model, the roles of the predecessor and successor gradually change, with the successor taking on more power while their predecessor withdraws. To begin (pre-succession, or phase 1), the predecessor is the sole operator/leader of the company. In this stage, one or more potential successors has been identified and is undergoing education and preliminary training, but is not yet heavily involved with the business and plays no formal leadership role. Over time (training, or Phase 2), the chosen successor begins to undergo more formal training and act in a helper role, but does not yet hold real power in the organization. The predecessor takes on a 'monarch' role, with full control over decisions, while the successor acts as a 'helper'. In the transition phase (or Phase 3), the successor takes on more power and responsibility, while still gaining more in-depth knowledge. The predecessor's transition to an overseer and delegator role is paired with the transition of the successor to a managerial role, wielding some power but looking to the predecessor for overall decisions. The final stage of the succession process (Phase 4, or succession) is the transfer of organizational power and control to the successor, who now acts as the chief decision-maker. The predecessor then recedes into the background as an advisor or consultant to the successor [24,27]. Thus, this process is not just a gradual transfer of power and responsibility for decision-making, but also a process of successor learning, through which they are prepared for their eventual role.

*2.4. Knowledge Management's Role in the Succession Process*

The succession period is a challenging time for knowledge management within the firm, because it is a time when knowledge may be lost due to ineffective transfer to the successor [28]. Bracci and Vagnoni proposed a theoretical model by which the required knowledge transfer and creation processes for effective succession were dependent on the successor, incumbent and organization characteristics. These knowledge transfer and creation processes are dependent on social connections between the successor and incumbent, as well as external and organizational processes [28]. Another key finding on knowledge management in the succession process is that effective knowledge management for succession can actually improve the organization's knowledge creation and absorptive capacity [29]. Furthermore, knowledge management, particularly knowledge transfer, is essential for strategic renewal of the organization between generations [30]. Strategic renewal is a critical process in the family firm, as it allows the firm to maintain its relevance through multiple generations of leadership. The knowledge management process, through which existing knowledge is both passed on and transformed by newer generations, is one of the key processes by which the firm maintains its longevity [30].

Thus, these studies' findings support the importance of knowledge management in the succession process.

There have been several studies which have investigated the knowledge transfer process in family firms and the role it plays in the succession process [31–35]. In one case, which featured a family-owned high-tech measurement solution manufacturer, the firm developed what the authors termed 'idiosyncratic capabilities' to overcome the boundaries and barriers to intergenerational knowledge transfer for succession [31]. A multiple case study identified the concept of 'knowledge-creating human capital', which is a tool for knowledge transfer to facilitate succession by the gradual development of the knowledge of individuals in the line of succession [32]. A third study noted a specific barrier, that of cross-generational bias, and explored how it influenced cross-generational knowledge flows and the impact this had on the succession process [33]. A theoretical review identified a range of factors that influenced knowledge transfer in the succession process of family firms including trust, commitment to the business, intergenerational and intragenerational relationships, and other factors [34]. Finally, a study identified the role of power in intergenerational knowledge transfers and how it affects the succession process [36]. These studies



have identified several different knowledge creation and transfer processes, which could take place during the process of succession. These studies clearly show that knowledge creation and transfer processes are an inherent part of the succession process, and that knowledge transfer occurs throughout the process. However, one aspect of the literature which is missing is that few of these studies have clearly identified at what stage of the succession process these knowledge processes may take place. This was part of the model development process, which was undertaken to establish what kinds of activities took place and at what stage.

There are limits to the role of knowledge management in family firm succession. As one study points out, family firms cannot remain successful in the long term simply by passing down knowledge; knowledge transformation, or the development of innovation from existing knowledge, is one of the processes for long-term success of the family firm [33]. The implication for this study is that a simple transfer of knowledge is not enough to guarantee succession or the continued existence of the firm afterward. Furthermore, many firms do not make use of all possible resources for knowledge transfer, due to an emphasis on tradition rather than innovation and on 'familyness' rather than change over time [34]. Furthermore, family firms, especially smaller family firms, may not have a clear knowledge management process in place at all, which limits their potential for knowledge transfer in the succession process [34]. Moreover, there are broader organizational questions of trust and identity that will influence the organization's behavior as a whole, for example in domains like environmental responsibility [35], financial performance and investment outcomes [37]. This means that for many family firms, there can be severe limitations on both the use of knowledge management in the firm and the use of knowledge management for succession. The model developed here does try to acknowledge these limitations, particularly taking into account the limitations due to the resources and capabilities that smaller firms face [34].

*2.5. Theoretical Framework*

The theoretical framework (Figure 1) is based on Handler's [24,27] model of family firm succession, along with the SECI model of knowledge transfer [8,10]. An additional set of possible knowledge creation and knowledge transfer processes is also incorporated, each of which has been identified as a possible process for succession management.

In summary, although there are disparate models that address the firm's succession process [24,27] and knowledge management processes [8,10], there has not previously been an attempt to understand how knowledge management processes are used within the succession process. This research integrates the two models, along with knowledge creation and transfer processes identified as relevant to an effective succession in the family firm. This theoretical framework serves as the basis for a qualitative investigation of family firms in Thailand.

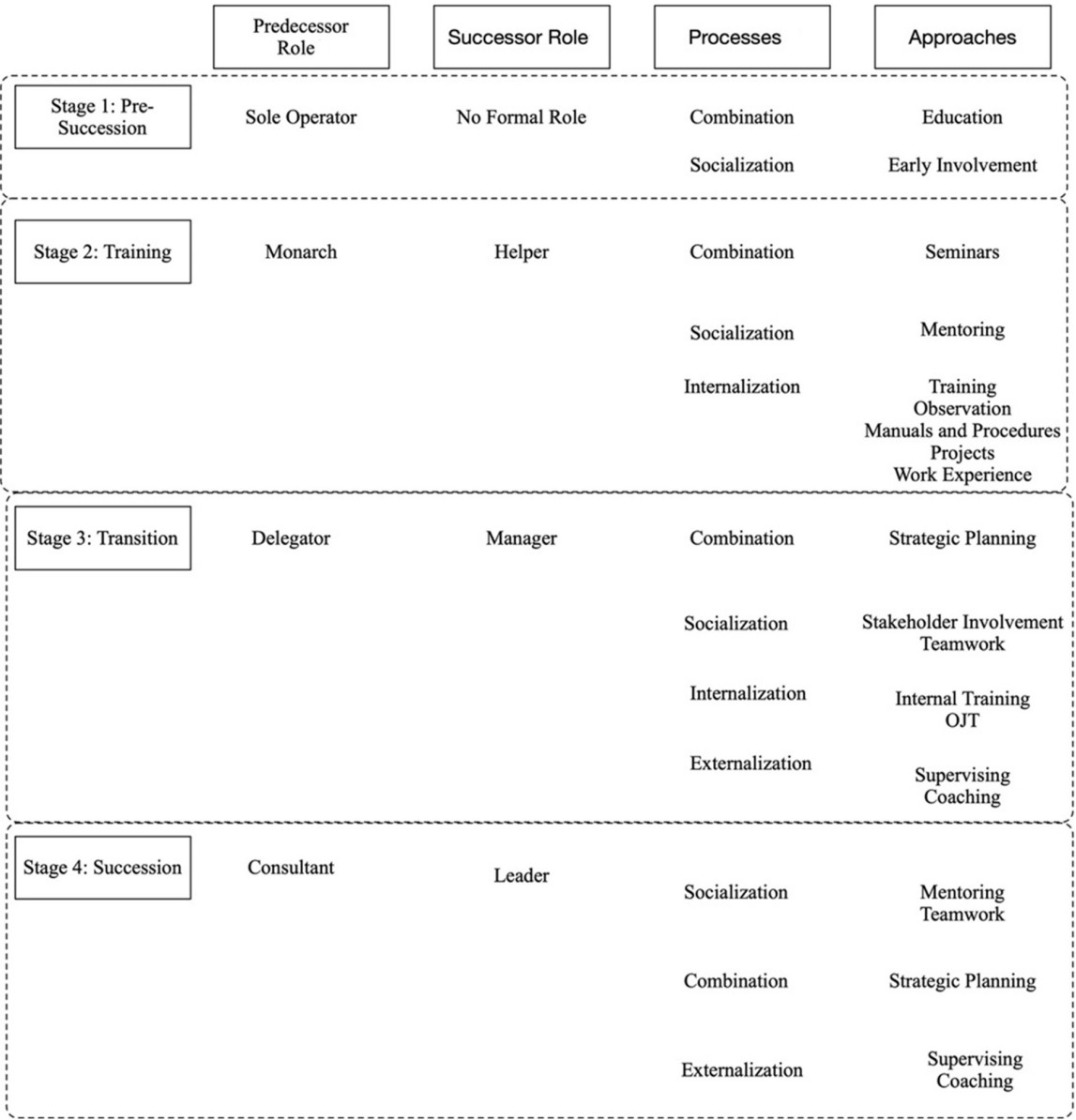

**Figure 1.** Theoretical framework of the paper.

### 3. Methodology

The study used an exploratory, qualitative multiple case study design, which was intended to allow for the most possible elaboration and exploration of the process of succession and the role of knowledge transfer within it. Exploratory qualitative research is intended to investigate phenomena in significant detail, allowing for exactly this process of fine-grained analysis [38]. The multiple case study approach was selected because it allows for in-depth analysis of specific instances and comparison between them [39]. This information-rich, highly detailed and specific design was chosen in recognition of the significant gap in the research outlined above.

Analysis was at the individual level, including perspectives of recently appointed and prior firm leaders of Thai family firms. A total of 30 firms were included (10 each of small, medium and large firms depending on the official Thai definition of firm size [39]). All firms were family firms, defined as those where more than 50% of economic ownership and/or significant managerial control was assigned to members of a single family [3]. Furthermore, all firms had undergone a generational succession process in the top leadership role within the past five years. A summary of the firms and interviewees is provided in Table 1.

**Table 1.** Summary of firms and participants [1].

| Firm Number | General Firm Operations Area | Predecessor | | Successor | |
|---|---|---|---|---|---|
| | | Family Position | Code | Family Position | Code |
| Small Firms | | | | | |
| 1 | Manufacturing | Father | 1P | Son | 1S |
| 2 | Retail | Father | 2P | Daughter | 2S |
| 3 | Personal Services | Father | 3P | Daughter | 3S |
| 4 | Manufacturing | Mother | 4P | Daughter | 4S |
| 5 | Food | Mother | 5P | Son | 5S |
| 6 | Manufacturing | Grandmother | 6P | Grandson | 6S |
| 7 | Hospitality | Mother | 7P | Daughter | 7S |
| 8 | Retail | Father | 8P | Son | 8S |
| 9 | Retail | Father | 9P | Son | 9S |
| 10 | Food Processing/Production | Father | 10P | Son | 10S |
| 11 | Food Processing/Production | Grandfather | 11P | Grandson | 11S |
| 12 | Food Processing/Production | Older Cousin | 12P | Younger Cousin | 12S |
| 13 | Food Service | Mother | 13P | Son | 13S |
| Medium Firms | | | | | |
| 14 | Food Service | Mother | 14P | Daughter | 14S |
| 15 | Manufacturing | Father | 15P | Son | 15S |
| 16 | Information Technology | Grandfather | 16P | Granddaughter | 16S |
| 17 | Manufacturing | Mother | 17P | Daughter | 17S |
| 18 | Tourism Services | Uncle | 18P | Niece | 18S |
| 19 | Hospitality | Mother | 19P | Son | 19S |
| 20 | Manufacturing | Father | 20P | Son | 20S |
| Large Firms | | | | | |
| 21 | Manufacturing | Mother | 21P | Daughter | 21S |
| 22 | Manufacturing | Father | 22P | Son | 22S |
| 23 | Manufacturing | Uncle | 23P | Nephew | 23S |
| 24 | Retail | Father | 24P | Son | 24S |
| 25 | Media | Aunt | 25P | Niece | 25S |
| 26 | Information Technology | Grandfather | 26P | Grandson | 26S |
| 27 | Personal Services | Father | 27P | Son | 27S |
| 28 | Hospitality | Grandfather | 28P | Grandson | 28S |
| 29 | Food Processing/Production | Father | 29P | Son | 29S |
| 30 | Food Service | Father | 30P | Daughter | 30S |

[1] Source: Author.

Data was collected using semi-structured interviews, which allowed the researcher to combine conciseness with the opportunity to explore the succession process in more detail and leaving space for participants to provide new information and challenge assumptions [40]. Interviews were conducted with the current and prior leader of each firm (the successors and predecessors), for a total of 60 interviews. The interviews were recorded and then transcribed for analysis. In some firms, additional data was also collected through observation (work shadowing with successor leaders) and secondary data provided by the firm, for example summary financial statements. However, this supplementary data was not available for all firms.

Data analysis was conducted using the NVIVO computer-assisted qualitative data analysis software package. The coding was performed using an iterative approach. The initial codebook was generated from the existing literature on knowledge transfer and succession planning [41]. The interviews were then coded, with successor and predecessor interviews from each firm coded together in order of collection. New codes were added to the codebook as needed. The initial coding process continued to the point of theoretical saturation, where no new codes were identified during the process [41]. Following this point, the codebook was finalized. The interviews were then coded again in the same order, with the new codebook applied. The results were tabulated and a narrative interpretation of the findings was prepared. The final stage of analysis was the development of the

succession model, in which the initial theoretical framework was adapted based on the findings of the research.

## 4. Findings

### 4.1. Knowledge Creation and Transfer Approaches

The interviews identified a total of 16 different knowledge creation and transfer approaches, through which the successors either received knowledge directly from their predecessors or other senior workers within the firm or developed it through social interaction and discovery. These processes (summarized in Table 2) were identified by other predecessors and successors as the most important aspects of the preparation process for succession. There were a few unusual approaches identified by only one or a few organizations (for example, self-led learning, as identified by one predecessor but not confirmed by their successor as relevant), but as these were uncommon and later analysis did not identify them as particularly important, they have been excluded from the overall model.

**Table 2.** A summary of the knowledge creation and transfer processes identified during the interviews [1].

| Knowledge Management Approach | Brief Definition | Illustrating Quotes | Frequency | |
|---|---|---|---|---|
| | | | # of Firms | % of Firms |
| Early Involvement | Involvement in the company prior to adult work life, for example weekend jobs. | *"I worked in the store during the weekends from when I was 13." (24S)* *"I began to introduce my daughter to my clients and suppliers in her teens." (14S)* | 55 | 91.7 |
| Education | Formal education related to managing the business, e.g., business or relevant technical degrees at the vocational or undergraduate level. | *"I studied tourism at university because I knew I would be taking over my aunt's business." (18S)* | 53 | 88.3 |
| Work Experience | Experience in work-related roles in the company or in other companies. | *"I started on the manufacturing line in high school, then moved into a supervisor role." (1S)* | 47 | 78.3 |
| Observation | Observational training in different roles in the company, including job shadowing and internships. | *"I spent my summers in university shadowing the design, production and sales departments." (21S)* | 40 | 66.7 |
| Seminars and Courses | External, short-term non-university training related to the business and its activities, including technical and non-technical information. | *"I have done my food safety certification and first aid training already." (30S)* | 36 | 60 |
| Mentoring | A close formal or informal relationship with one or more older employees to assist in organizational and technical problems and make social connections. | *"The plant supervisor was my informal mentor, he taught me how the process worked and introduced me to a lot of important people." (17S)* | 58 | 96.7 |
| Apprenticeship | A formal period of training, typically in technical or vocational roles, either inside or outside the company. | *"I did a computer science apprenticeship in university which helped me understand the bigger technology picture." (16S)* | 36 | 60 |

**Table 2.** *Cont.*

| Knowledge Management Approach | Brief Definition | Illustrating Quotes | Frequency | |
|---|---|---|---|---|
| | | | # of Firms | % of Firms |
| Studying Manuals and Procedures | Formal or informal reading and learning of company policies and procedures, technical specifications and other information. | *"I learned almost everything about our manufacturing process by reading the manuals." (22P)* | 38 | 63.3 |
| Project Work/ Problem Solving | Involvement in projects and solving problems within the company, either as a team member or a team leader. | *"I am on the company's customer service response team, which deals with customer service failures, online complaints and other problems. Our goal is to make the customer happy even if we failed the first time." (19S)* | 51 | 85 |
| Internal Training | Participation in formal training programs in the company (provided internally or through external programs), typically to learn different technical and organizational roles. | *"Before taking over the company I was expected to complete our internal management training course. My grandson is doing the same." (28P)* | 44 | 73.3 |
| On-The-Job Training (learning by doing) | Participation in informal or OJT training programs related to different organizational roles (commonly lower-level roles such as manufacturing). | *"My first learning experiences were OJT from the line supervisor and line workers." (20P)* | 60 | 100 |
| Teamwork | Engaging with teams in the company, both as team members and team leaders, for short-term projects or long-term problem solving. | *"Right now I lead a technical team, we are investigating upgrading the manufacturing line to Industry 4.0." (22S)* | 60 | 100 |
| Stakeholder Involvement | Playing a significant role in stakeholder management, including customers, suppliers, and others, to achieve the company's strategic objectives. | *"I have been working with our logistics and supply chain department for a few years now, dealing with our suppliers." (24S)* | 55 | 91.7 |
| Supervising | Supervising and managing other employees in formal teams or departments. | *"Right now, I manage one of our locations." (13S)* | 60 | 100 |
| Coaching | Participating in coaching relationships with other employees, either as a coached employee or coach, to improve performance. | *"I work with our new graduate training program, and have coaching sessions with our interns." (24S)* | 60 | 100 |
| Strategic Planning | Participating in the strategic planning process for the firm, as team member or team leader. | *"I have been involved in our strategic planning process for about five years. It's the last thing my father has kept control of." (29S)* | 48 | 80 |

[1] Source: Author.

The frequency of these knowledge creation and transfer processes (also summarized in Table 2) varied somewhat, though there was some consistency in the processes used. There were a few approaches, including OJT, teamwork, supervising and coaching, which were used in all firms. Several others, including early involvement, education, mentoring, project work and problem solving, and stakeholder involvement were used in at least five of six firms. At least two of three firms used work experience, observation, internal training and strategic planning involvement. The least frequently used approaches included seminars

and courses, apprenticeships, and studying manuals and procedures, which were used by more than half of the firms but less than two-thirds. Thus, although there is a high degree of similarity in knowledge creation and transfer processes, it is not entirely consistent between firms.

### 4.2. Knowledge Forms and Knowledge Processes

Following the identification of the knowledge creation and transfer processes above, the researcher investigated what forms of knowledge they were applied to (tacit or explicit knowledge) and which of the SECI knowledge transformation processes they were used within (Table 3). This follows the SECI model of knowledge transformation and the relationship of tacit and explicit knowledge [8,10].

**Table 3.** Forms and transformations of the succession knowledge processes [1].

| Knowledge Process | Knowledge Forms | | Used in SECI Processes of ... | | | |
| --- | --- | --- | --- | --- | --- | --- |
| | Tacit | Explicit | Socialization (T-T) | Externalization (T-E) | Combination (E-E) | Internalization (E-T) |
| Early Involvement | ✓ | ✓ | ✓ | | | ✓ |
| Education | | ✓ | | | ✓ | ✓ |
| Work Experience | ✓ | ✓ | ✓ | | ✓ | ✓ |
| Observation | ✓ | | ✓ | | | |
| Seminars and Courses | | ✓ | | | ✓ | |
| Mentoring | ✓ | ✓ | ✓ | ✓ | | ✓ |
| Apprenticeship | | ✓ | | | ✓ | ✓ |
| Studying Manuals and Procedures | | ✓ | | | ✓ | ✓ |
| Project Work | ✓ | ✓ | ✓ | ✓ | ✓ | ✓ |
| Internal Training | ✓ | ✓ | | ✓ | | ✓ |
| OJT | ✓ | ✓ | ✓ | ✓ | | ✓ |
| Teamwork | ✓ | | ✓ | | | |
| Stakeholder Involvement | ✓ | | ✓ | | | |
| Supervising | ✓ | ✓ | | ✓ | | ✓ |
| Coaching | ✓ | ✓ | ✓ | ✓ | | ✓ |
| Strategic Planning | | ✓ | | | ✓ | |

[1] Author, based on the SECI model of Nonaka and Takeuchi [10].

Each of these knowledge processes was configured differently. Processes that were oriented toward only tacit knowledge included observation, teamwork and stakeholder involvement, while explicit knowledge-oriented processes included education, seminars and courses, apprenticeship, studying manuals and procedures, and strategic planning. Processes that combined tacit and explicit knowledge in various ways included early involvement, work experience, mentoring, project work, internal training, OJT, supervising, and coaching. Thus, half of the approaches that were identified were associated with both tacit and explicit knowledge. Furthermore, as summarized in Table 3, most of these processes could be associated with more than one of the SECI model's knowledge transformation processes. Only observation, seminars and courses, teamwork and stakeholder involvement seemed to be aligned to only one of these processes. From this analysis, it is possible to see that the knowledge creation and transfer processes used in succession planning and preparation do not serve only a single purpose. Instead, there is a dynamic interaction between tacit and explicit knowledge for many of them, and this dynamic interaction enables more than one kind of knowledge transformation.

*4.3. Knowledge Processes across the Succession Phases*

Following the analysis of the knowledge processes based on the SECI model above, the next stage was to investigate how these knowledge processes occurred across the phases of succession. Handler's succession framework has four stages, including pre-succession, training, transition and succession [24,27]. This framework was used as the basis for the interviews. However, while participants clearly identified the pre-succession and succession phases of transition, there was no clear differentiation between the training and transition phases. Instead, this was viewed by participants as one long stage of preparation and training, during which the successor gradually took over more responsibility from their predecessor. Thus, the analysis of knowledge creation and transfer processes followed the insight of the participants, collapsing phases 2 and 3 of Handler's succession framework into a single Transition phase [24,27]. Table 4 summarizes the knowledge processes used across these three succession phases.

**Table 4.** Knowledge creation and transfer processes throughout the succession phases.

| Knowledge Approach | Phase 1: Pre-Transition | | Phase 2: Transition | | Phase 3: Succession | |
|---|---|---|---|---|---|---|
| | Total | % | Total | % | Total | % |
| Early Involvement | 55 | 91.7 | 0 | 0.0 | 0 | 0.0 |
| Education | 53 | 88.3 | 0 | 0.0 | 0 | 0.0 |
| Working Experience | 47 | 78.3 | 0 | 0.0 | 0 | 0.0 |
| Observation | 20 | 33.3 | 40 | 66.7 | 0 | 0.0 |
| Seminars and Courses | 3 | 5.0 | 12 | 20.0 | 21 | 35.0 |
| Mentoring | 55 | 91.7 | 58 | 96.7 | 58 | 96.7 |
| Apprenticeship | 36 | 60.0 | 6 | 10.0 | 0 | 0.0 |
| Studying Manuals and Procedures | 0 | 0.0 | 38 | 63.3 | 0 | 0.0 |
| Project Work | 0 | 0.0 | 51 | 85.0 | 0 | 0.0 |
| Internal Training | 34 | 56.7 | 44 | 73.3 | 0 | 0.0 |
| OJT | 35 | 58.3 | 60 | 100.0 | 0 | 0.0 |
| Teamwork | 50 | 83.3 | 60 | 100.0 | 35 | 58.3 |
| Stakeholder Involvement | 26 | 43.3 | 55 | 91.7 | 42 | 70.0 |
| Supervising | 43 | 71.7 | 60 | 100.0 | 60 | 100.0 |
| Coaching | 46 | 76.7 | 60 | 100.0 | 22 | 36.7 |
| Strategic Planning | 2 | 3.3 | 40 | 66.7 | 48 | 80.0 |

In Phase 1 (Pre-transition), there were several activities that more than half of respondents identified, including early involvement (91.7% of firms), education (88.3%), work experience (78.3%), mentoring (91.7%), apprenticeship (60%), internal training (56.7%), OJT (58.3%), teamwork (83.3%), supervising (71.7%) and coaching (76.7%). Thus, during this phase, there is already active participation of the successor, including involvement in the company's activities and work live. However, there are some important differences between this phase and later phases. For example, the successors' role in activities like mentoring, teamwork, and coaching were more likely to be as a follower, coached employee or mentee rather than in the leadership role. For example, 13P entered the company as an ordinary team member, with her mother acting as her supervisor, a role she reprised for her son. It was also common at this stage for employees to undergo training, including formal training and OJT, as part of the company's ordinary training process before moving onto a team as a normal worker, rather than immediately taking on a powerful role in the company. Thus, during the Pre-transition phase, while the successor is already taking part in formal knowledge creation and transfer activities and playing a role in the company, they do not yet have a role of power or control within the organization. This indicates that, rather than using entitlement nepotism to hire potential successors into roles they are not yet prepared for, most of the firms are actually using reciprocal nepotism, with successors using the pre-transition time to form social relationships and create networks of support and obligation that assist in their knowledge creation and transfer [16]. Thus, even though

a family firm is by nature nepotistic, it appears that, at least for successful firms, simply assigning successors roles of responsibility at an early stage does not occur.

Phase 2 (Transition) is marked by a transition away from some of the activities of Phase 1, while in others the successor takes on more senior roles. Activities including early involvement, education, and work experience are no longer part of the training process. However, activities like observation (66.7% of firms), mentoring (96.7%), studying manuals and procedures (73.3%), project work (85%), internal training (73.3%), OJT (100%), teamwork (100%), stakeholder involvement (91.7%), supervising (100%), coaching (100%) and strategic planning involvement (66.7%) are now commonly used by successors to create and transfer knowledge. In some cases, these activities continue as before. For example, internal training and studying manuals and procedures may be used by successors to better understand the company's activities, including roles they have not worked in previously. In other cases, the knowledge creation and transfer activities are changing during this phase. Successors take on supervisory roles, rather than being supervised, and are more likely to act as mentors and coaches for junior employees rather than being coached and mentored. They are also more likely to take on leadership roles, such as teamwork and leading projects and problem-solving activities. However, they also still continue to have knowledge transferred to them by their senior mentors and predecessors. For example, 17S reported that, as he began to meet with customers, his parents mentored him, passing on tacit knowledge about the customers and their previous experiences with them. Thus, even though they are taking on senior roles, they are still learning through a process of social interaction.

At this stage, the successors also become much more involved in activities that involve social interactions with external organizational partners, for example being involved with stakeholder management activities and strategic planning activities. Thus, this is a long stage that is poorly differentiated, but which does involve a gradual process of both accumulation of power and responsibility [24,27] and changes in the knowledge creation and leadership processes.

In Phase 3 (Succession), successors are still engaged in knowledge creation and transfer processes to some extent. Most of the successors and predecessors reported that mentorship relationships continued into this stage (96.7%), consistent with the role of the predecessor in the succession framework [24,27]. Teamwork (58.3%) and stakeholder involvement (70%) were still commonplace activities as well. In the other common activities, including supervising (100%) and strategic planning (80%), successors were more likely to engage in supervision and leadership roles. However, they were still engaging with both followers and their predecessors, who continued to provide useful knowledge, especially tacit knowledge about business partners and strategy. Thus, while the Successor phase seems to be a less intense period of learning, it remains a period of learning and knowledge transfer.

### 4.4. Knowledge Processes in Different Firm Sizes

The final aspect of development for the knowledge process model for succession was an investigation of how knowledge processes were different in firms of different sizes. The frequency of knowledge processes in firms of different sizes is summarized in Table 5. While about half the knowledge processes were common among firms of all sizes, there was one process more common in small firms and seven that were more common in medium and large firms.

**Table 5.** Knowledge creation and transfer processes in firms of different sizes [1].

| Knowledge Creation/ Transfer Approach | Small Firms | | Small Firms | | Large Firms | |
|---|---|---|---|---|---|---|
| | Total | % | Total | % | Total | % |
| Early Involvement | 20 | 33.3 | 20 | 33.33 | 15 | 25.0 |
| Education | 14 | 23.3 | 19 | 31.7 | 20 | 33.3 |
| Working Experience | 11 | 18.3 | 17 | 28.3 | 19 | 31.7 |
| Observation | 4 | 6.7 | 16 | 26.7 | 20 | 33.3 |
| Seminars and Courses | 6 | 10.0 | 10 | 16.7 | 20 | 33.3 |
| Mentoring | 20 | 33.3 | 18 | 30.0 | 20 | 33.3 |
| Apprenticeship | 18 | 30.0 | 10 | 16.7 | 8 | 13.3 |
| Studying Manuals and Procedures | 6 | 10.0 | 14 | 23.3 | 18 | 30.0 |
| Project Work/Problem Solving | 18 | 30.0 | 13 | 21.7 | 20 | 33.3 |
| Internal Training | 8 | 13.3 | 16 | 26.7 | 20 | 33.3 |
| OJT | 20 | 33.3 | 20 | 33.3 | 20 | 33.3 |
| Teamwork | 20 | 33.3 | 20 | 33.3 | 20 | 33.3 |
| Stakeholder Involvement | 20 | 33.3 | 17 | 28.3 | 18 | 30.0 |
| Supervising | 20 | 33.3 | 20 | 33.3 | 20 | 33.3 |
| Coaching | 20 | 33.3 | 20 | 33.3 | 20 | 33.3 |
| Strategic Planning | 11 | 183 | 17 | 28.3 | 20 | 33.3 |

[1] Source: Author.

### 4.4.1. Shared Knowledge Processes

Some of the knowledge processes were observed at around the same frequency in small, medium and large firms. These shared processes included early involvement, mentoring, project work and problem solving, OJT, teamwork, stakeholder involvement, supervision and coaching.

### 4.4.2. Processes More Common in Small Firms

Apprenticeships were common in small firms. However, they were much less common for medium and large firms.

### 4.4.3. Processes More Common in Medium and Large Firms

There were several knowledge processes that were far more common in medium and large firms than they were in small firms. These activities included work experience, observation, internal training, seminars and courses, and studying manuals and procedures, along with strategic planning involvement. Between 70% and 100% of medium and large firms used these processes, but typically 55% or less of small firms. Education, used in almost all medium and large firms, was slightly less common in small firms, though this difference was not as sharp as some of the other processes.

The reason for these differences can be understood through the lens of knowledge management processes of small firms in general, which are affected by resource constraints. For example, small firms' more frequent use of apprenticeships can be understood as a way to access external knowledge from other firms, which is a common knowledge strategy for small firms [21]. Specifically, many of the apprenticeships discussed in the interviews were not with the organization itself, but externally arranged at competing firms or in other industries. Thus, the use of apprenticeships in preparation of the successor means that the firm can also draw on the knowledge resources of other organizations. This could ultimately increase the firm's absorptive capacity [23], overcoming one of the key constraints of the small firm and enabling higher levels of knowledge creation and innovation in the future. There is also some possibility that resource and capacity constraints limit the potential for successors in small firms to make full use of the organizational resources available [22]. For example, 7P observed during the interviews that activities like observation and job shadowing were not possible because the organization simply did not have the spare capacity for the successor to engage in internships or similar activities. Resource constraints like low levels of process documentation and lack of formal training procedures also meant

that some of the knowledge transfer activities were less common or simply not used in small firms. Thus, there are several differences in the resources and capacities of small firms and those of larger firms that changed how successors were prepared, particularly in the pre-transition stage, compared to the medium and large firms. However, small firms did not show more reliance on socialization processes in general as presupposed in some of the literature [21]; instead, many of the socialization processes were used by firms of all sizes. Perhaps a better explanation is that small firms are equally likely to engage in socialization and internalization processes, but less likely to engage in the explicit knowledge transfer processes of externalization and combination. One of the unexplained differences is the lower engagement with strategic planning activities. However, this was explained in the interviews, as several of the interviewees (1P, 3S, 4P, 4S, 5P, 5S, 7P, 8S, 9P) indicated that the company simply did not do a lot of strategic planning. Thus, this is a substantive difference in the operations of small firms compared to medium and large ones.

Another major difference between small firms and medium and large firms was that successors took on roles with significant responsibilities earlier in the succession period. For example, it was more common for successors in small firms to take on teamwork, supervision, and strategic planning and stakeholder involvement roles during the pre-transition phase, while successors in large firms were more likely to still be engaged in further education and formal training processes. Elaboration from small firm participants made it clear that successors were expected to take on more responsibility earlier than those in larger firms, and needed to either spend time learning technical skills through apprenticeship or move directly to a leadership role. Although this may look like a form of entitlement nepotism [16], which would otherwise be unhelpful to the succession process, the interviews revealed that this was more about resource constraints. Simply, as 6S stated, there was no room in the small firm (with fewer than 10 employees) for the successor to engage in a protracted learning period, due to resource constraints that limited non-productive employees. This contrasts sharply with the experience of large firm successors like 27S, who spent nearly a decade learning positions in the company from the ground up.

*4.5. Knowledge Process Model of Family Firm Succession*

The final stage of the research was to develop a knowledge process model of family firm succession (Figure 2), taking into account the knowledge processes identified, the phases during which these processes occur, and the differences between small, medium and large firms. The model developed includes the activities reported by at least 75% of participants, either all participants (the shared knowledge processes) or within a firm size category (the firm size-dependent knowledge processes). The derived process model addresses, therefore, two dimensions of the ba or knowledge context [6]: the firm size, which affects both the internal knowledge processes of the firm and its resource and capacity constraints; and the stage of succession, which affects the knowledge processes and capabilities of the successor. As the model also shows, the SECI processes occur throughout all three stages of the succession process; thus, there is little sense that any particular process predominates at any given stage, although explicit knowledge processes are more common in Phases 1 and 2.

This model is a preliminary model and designed for further research into the knowledge creation and transfer processes that occur in the family firm succession process. The next steps for the development of the research model is a broader application to investigate knowledge creation and transfer in other conditions, including outside the initial sample, for failed or difficult succession processes, and in cross-cultural contexts (including different countries and different Thai cultures).

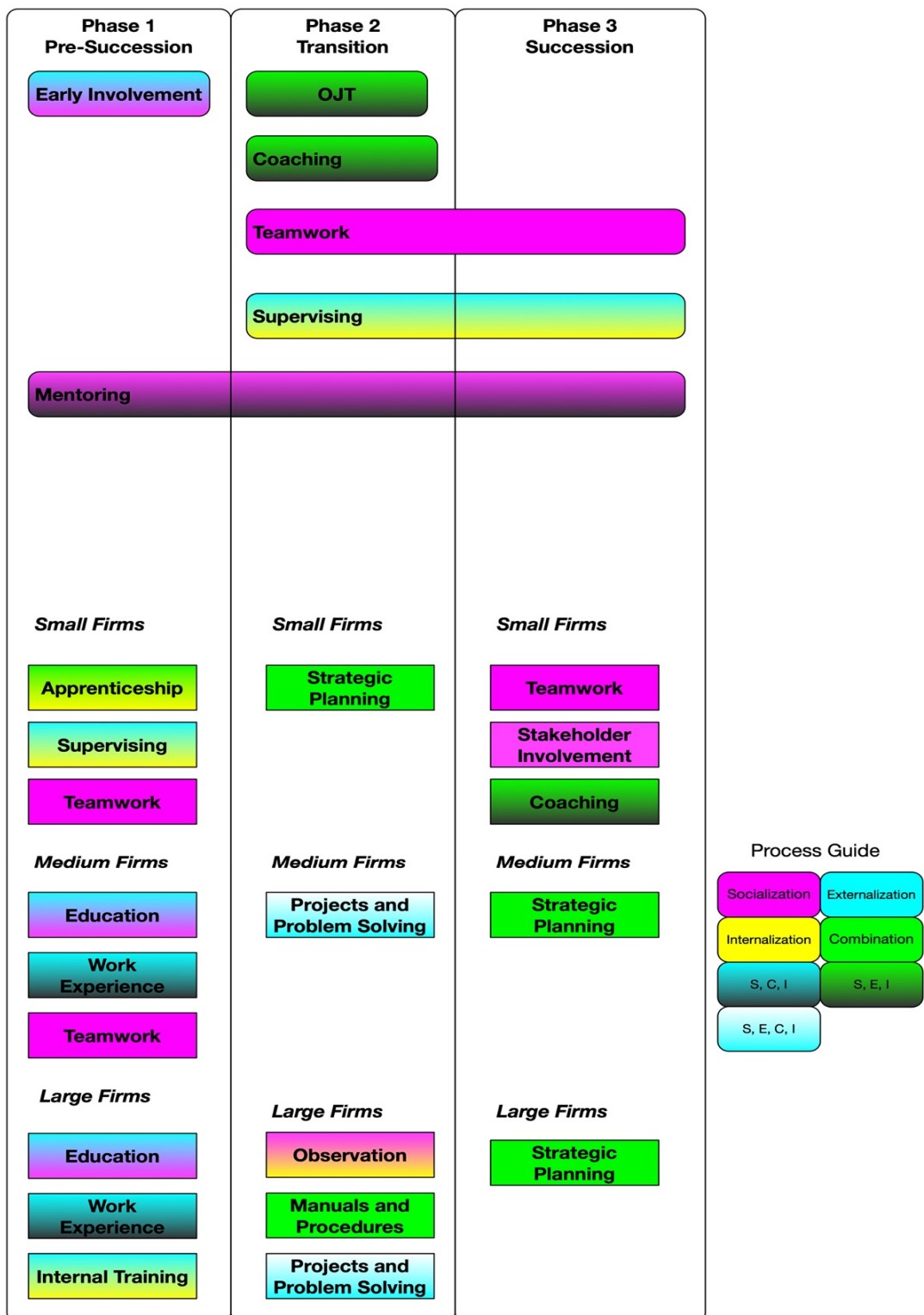

**Figure 2.** A knowledge process model of family firm succession.

*4.6. Discussion*

There are some implications of the literature for the model as developed here. One of these implications is that relationships within the family surround and influence the processes here. For example, the nepotism-based relationships of the family firm [16] may

mean that planned or potential successors have access to training opportunities that do not ordinarily exist. An example is the early involvement of planned successors in the operations of small family firms, which would not be available to non-family members. These positions may be driven partly by tradition, which holds greater sway than it does in non-family firms [17]. At the same time, the relationships of trust and affective bonds between family members will also influence the extent to which individuals have access to these training opportunities [19,20]. Successors and predecessors who have strong bonds of trust and affection may, therefore, be more effective at transferring knowledge than those whose relationship is less close. There are also other issues that could influence outcomes, including intergenerational biases, commitment to the organization and its success, and flows of power [33–35]. Therefore, the processes of succession can be expected to look different from firm to firm, and between generations within the same firm. There may also be competing tensions, for example the effect of tradition, which can limit innovation [18] against the demand for strategic renewal [30]. While firms may develop ways around these tensions to ensure effective knowledge transfer in any case [31,32], this is not guaranteed.

## 5. Conclusions

### 5.1. Academic Implications

The academic contribution of this research lies in its integrated process model of knowledge creation and transfer for the succession process. This model, which incorporated the concepts of the SECI model along with the succession phase model of the family firm's lifecycle and the firm size and knowledge processes identified within the research, is a novel approach to understanding knowledge creation and knowledge transfer during the succession process. The key insight of the model is that knowledge transfer for succession in the family firm is not a one-time event. Instead, it is a continual process in which the successor's knowledge gradually becomes more specific and more focused on higher-end issues such as leadership and strategy. Furthermore, the different resources and constraints in firms of different sizes means their knowledge process is different throughout the succession lifecycle. Thus, it is not possible to speak of knowledge processes for succession as a single set of practices—instead, these practices are heavily dependent on the organizational environment and context. This model offers other researchers a tool for understanding knowledge transfer throughout the lifecycle of the family firm, as well as insight into what kinds of processes may be used.

### 5.2. Managerial Implications

This research has some key findings that are relevant to knowledge creation and transfer processes in the family firm succession process, and which can be used to improve the management succession planning for the firm. First, the research has revealed that the knowledge creation and transfer processes of the family firm's succession span the length of the succession process, beginning long before the chosen successor takes the reins and continuing after their formal succession period. This implies that the succession process should not only consider knowledge transfer at the point where a successor is named, but instead potential candidates should be included in knowledge transfer well before this point. Furthermore, the 16 knowledge processes identified are not used indiscriminately throughout the succession process. Instead, the pre-succession stage is characterized by external, formal learning processes and experience- and knowledge-building activities such as early firm experience. During and after the succession process, knowledge becomes more internal to the firm, and, to an extent less formalized, the chosen successor also begins to gain knowledge through formal involvement in the firm's strategy, planning and operations, rather than through external learning. Thus, the candidates for succession should all undergo the initial, formal training process.

This research also revealed that that firms of different sizes have different succession knowledge processes. Successors in small firms, with their informal knowledge processes and resource constraints, rely heavily on external activities like apprenticeships but, con-

versely, may be less driven to formal education and training. In large firms, where there are more resources but, simultaneously, more demands on the successor, formal education and training is prevalent. Medium-sized firms fall in the middle, with some processes more in common with small firms and others consistent with large firms. Thus, while knowledge creation and transfer is part of the succession process for firms of all sizes, exactly how this takes place depends on the firm size, which is a proxy for resources and internal process formalization. Therefore, there is no one-size-fits-all solution for knowledge transfer within the succession process, and firms need to make choices that effectively prepare potential successors within the resources available to the firm.

*5.3. Limitations*

There are some limitations to this research, which suggest opportunities for further research. One of these limitations is that the extent of cross-cultural generalizability of the knowledge process succession model is unclear. Family firm succession processes do vary widely between cultures, and there are known issues in generalizability of the SECI model, especially as it relates to tacit knowledge and its transformation. Given these empirical and theoretical differences, it is possible that the research model may not generalize to all cultures. The second limitation is that the study only drew on the success cases of intergenerational transfer—failure cases were not investigated. Thus, it is unknown whether the factors identified are characteristic of success and not failure, or whether some or all of these conditions may also be in place in failed successions.

*5.4. Opportunities for Further Research*

There are several opportunities for further research that can be suggested, given the limitations of the current study. The issue of cross-cultural generalization of the findings can be addressed by cross-cultural model testing, in which the derived model is applied to firms in different cultures and explicitly investigated in relation to cultural differences. To address the limitation of positive bias, which occurred because this study only addressed success cases, it would be appropriate to incorporate failure cases, either through an extended case study or by including firms that failed in the transition period in a broader survey of family firms, when testing the model.

**Author Contributions:** Conceptualization, K.P. and V.V.R.; Data curation, K.P. and V.V.R.; Formal analysis, K.P. and V.V.R.; Investigation, K.P. and V.V.R.; Methodology, K.P. and V.V.R.; Project administration, K.P. and V.V.R.; Supervision, V.V.R.; Writing—original draft, K.P. and V.V.R. All authors have read and agreed to the published version of the manuscript.

**Funding:** This research was funded by The Royal Golden Jubilee Ph.D. Programme grant number [PHD/0075/2558].

**Conflicts of Interest:** The authors declare no conflict of interest.

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
