# Peer review of "The Role of Knowledge Creation and Transfer in Family Firm Succession"

_sustainability, doi:10.3390/su14105845_

Round 1
Reviewer 1 Report
The article is interesting, and the researched problem has scientific potential. However, some problems need to be solved:
1. Literature review should include more recent sources (2018-2021) and must be enriched with relevant references.
2. In the paper, I did not find the research questions, the research gap, and the hypotheses, which cannot be missing in well-founded research. Therefore, the author should provide a research design (a graphical representation) in the methodology section.
3. The findings and discussion sections should be separated.
4. Data processing is performed using descriptive statistics. The article would gain value if complex statistical methods were used to establish the relationships between variables. NVIVO computer-assisted qualitative data analysis software package offers complex features: building relationships within NVivo, including creating socio-grams and analyzing quantified qualitative data.
5. Improve the coherence and strengthen your arguments by discussing the implications of those studies for your research.
6. In my opinion, a section of conclusions that includes delimitated theoretical and managerial implications, research limitations, and future research directions would be helpful.
The article presents some scientific value and can be published after carefully reviewing the reported issues
Author Response
Dear Reviewer,
We appreciate your precious time in providing valuable suggestions. It was your insightful comments that led to possible improvements in the current version. We have revised and tried our best to address every one of them. We hope the manuscript revision meet your standards. We also welcome further constructive comments if any. All modifications in the manuscript have been highlighted in yellow.
Please see the attachment.
Best Regards,
Authors

Reviewer 2 Report
The introduction is too general. The authors should shorten the introduction by better explaining the research gap, the research question, the way how the research question is transposed into the practical part of the paper, as well as to highlight how the paper contributes to knowledge enhancing. The novelty of the paper must also be shown.
The last paragraph of the introduction should contain a brief description of the next sections of the paper.
The lit review
The relationship between corporate social responsibility and financial behavior as regards knowledge creation and knowledge transfer processes has not been covered, and thus such recent sources should be cited: May, A. Y. C., Hao, G. S., and Carter, S. (2021). “Intertwining Corporate Social Responsibility, Employee Green Behavior and Environmental Sustainability: The Mediation Effect of Organizational Trust and Organizational Identity,” Economics, Management, and Financial Markets 16(2): 32–61. doi: 10.22381/emfm16220212. Priem, R. (2021). “An Exploratory Study on the Impact of the COVID-19 Confinement on the Financial Behavior of Individual Investors,” Economics, Management, and Financial Markets 16(3): 9–40. doi: 10.22381/emfm16320211. https://www.frontiersin.org/articles/10.3389/fenvs.2022.781075/full The lit review is quite well developed. But figure 1 is not clear at all. the methodology is properly developed. The results are presented nice Maybe more comparisons between own findings and previous results from the literature should be shown The conclusions must be structured as: theoretical implications; managerial contributions; limitations; future research perspectives No references should be cited in conclusions.Author Response
Dear Reviewer,
We appreciate your precious time in providing valuable suggestions. It was your insightful comments that led to possible improvements in the current version. We have revised our manuscript and tried our best to address every one of them. We hope the manuscript meet your standards. We also welcome further constructive comments if any. All modifications in the manuscript have been highlighted in yellow.
Please see the attachment.
Best Regards,
Authors

Round 2
Reviewer 1 Report
Dear authors,
Thank you for improving your paper.
I believe it can be published in current form.
Best regards,
Reviewer 2 Report
As the authors have implemented all suggestions, the paper can now be accepted.